# Sutureless Technique for Surgical Castration in Adult Boars: A Feasibility Study

**DOI:** 10.3390/ani13030407

**Published:** 2023-01-26

**Authors:** Stella Maria Teresa Romeo, Sarah Morrone, Toufic Akl, Antonio Scanu, Nicolò Columbano

**Affiliations:** Department of Veterinary Medicine, University of Sassari, 07100 Sassari, Italy

**Keywords:** boars, surgical castration, swine, orchiectomy, sutureless technique, feasibility, field conditions

## Abstract

**Simple Summary:**

The use of boars for breeding purposes is common on small farms in rural regions of Italy. These animals are butchered at the end of their reproductive life for the production of charcuterie. In this regard, orchiectomy is required to remove boar taint caused by testosterone metabolites. The purpose of this study is to validate the efficacy of a safe, rapid and inexpensive castration procedure. To accomplish this objective, 91 boars were orchiectomised using the suture ligation technique. Following castration, characteristics of the severed ductus deference were evaluated in order to understand how much force would be required to tie knots without breaking the deferens and achieving proper haemostasis. After identifying a correlation between tensile strength and the age and weight of boars, a sutureless technique was successfully applied to 20 animals.

**Abstract:**

The heterogeneity of Italian manufacturing processes results in the production of a large variety of pork products. In Sardinia, boars are raised and butchered to produce charcuterie. These animals are castrated before slaughter as androstenone would otherwise taint the meat, rendering it unfit for human consumption. However, to date, the literature concerning surgical orchiectomy in adult boars is limited. The goal of this study is to assess whether a sutureless swine orchiectomy procedure is feasible. Additionally, this study aims to determine the appropriate traction force needed to tie knots in the deferens duct of pigs of different weights and ages. Two groups were created: the first (n = 91) underwent orchiectomy by suture ligation; the second (n = 20) was castrated using the sutureless technique. Deferens ducts of animals in the first group (n = 182) were collected following castration, and their tensile strength was measured. Pearson’s linear correlation was used to determine the relationship between the maximum traction force and weight and age groups. A correlation of 0.99 and 0.96 was shown between traction force and age and traction force and weight, respectively. In accordance with these results, sutureless castration was performed on 20 animals, calibrating the pulling force needed according to the age and weight of the boars. No complications were observed during the feasibility study, thus validating sutureless orchiectomy in adult boars.

## 1. Introduction

Pork is the most consumed meat in the European Union (EU), and the pig sector is one of its most economically significant farming industries. Furthermore, the EU is the world’s second-largest producer of pork after China and the world’s largest exporter of pork products [1]. Germany, Spain, France, and Italy account for approximately 50% of the EU’s overall production [2]. According to the National Institute of Statistics (ISTAT), the Italian pork sector is valued at roughly EUR 1.7 billion, of which processed pork makes up 85% [3]. Ham, bacon, guanciale, and many other products are omnipresent in Italian traditional culinary culture, and some, like prosciutto di Parma (DOP) and prosciutto San Daniele (DOP), are exported all around the globe. The significance of pork products in Italy is reflected by the importance of its pig farming industry, consisting of numerous small farms commonly found in rural areas of the country. Sardinia, known for being one of Italy’s biggest agricultural areas, is one such region. In this context, the castration of mature boars is often required since farmers mainly rely on natural reproduction through mating. At the end of their reproductive term, such boars are castrated in order to avoid boar taint caused by the presence of androstenone making the meat unfit for human consumption [4,5]. Whereas the surgical castration of young piglets is a standard routine procedure, the castration of adult boars is more complex and less described [6]. The latter usually requires general anesthesia since large boars are rather dangerous to handle and restrain. More importantly, orchiectomy of adult boars entails the use of haemostatic measures such as the ligation of the spermatic cord with surgical sutures or the crushing of the cord with an emasculator. Both these actions involve the introduction of foreign objects to the surgical site, as well as excessive manipulation, increasing the risk of hemorrhage and surgical contamination and, therefore, infection and abscess, which are the most common complications of orchiectomy [7].

In light of this, a novel sutureless castration technique was devised that does not require the use of foreign objects. This technique was inspired by the castration method for tomcats and consists of an open castration where haemostasis is achieved by splitting the vas deferens from the testis and tying several simple knots using the deferens duct and the testicular vasculature. This study aims to assess the viability of this sutureless technique as well as identify possible correlations between animal age/weight and technical surgical effectiveness.

## 2. Materials and Methods

The study was conducted in Sardinia during activities of the Mobile Clinic organized by the Veterinary Medicine Department of the University of Sassari between October 2020 and December 2021. 

### 2.1. Animals

Ninety-one boars (group 1) were subjected to conventional orchiectomy by suture ligation. In order to determine the tensile strength of the ductus deferens according to the age and weight of the animals, 182 deferens were removed and measured. Following this, 20 boars (group 2) were castrated employing the newly developed sutureless technique. All animals included in the study were older than 12 months and had mated at least once. Procedures were performed on site (farms) after the preparation of a clean and sanitized ad hoc surgical area. Informed consent was acquired from all breeders for the performance of the novel sutureless procedure. 

### 2.2. Anesthesia

Identical anesthetic protocols were applied for all boars included in this study. Animals scheduled for castration were fasted for 12 h prior to the surgery and isolated from the rest of the pigs on the farm in order to facilitate capture and restraint at the time of the intervention. Boars were restrained using a snout snare passed caudally of the superior canine teeth and over the dorsal surface of the snout. Peripheral intravenous (IV) administration of sedative and anesthetic drugs was achieved through the auricular veins, as seen in Figure 1. The weight of the animals was measured using a professional electric scale with a hook (PCE-CS 500, Ziboni Technology S.r.l., Rogno (BG), Italy).

Sedative and anesthetic drugs were administered in a single IV bolus. Sedation was achieved with 1 mg/kg of Azaperone (Stresnil, Elanco S.p.A. Italia, Sesto Fiorentino (FI), Italy) and 0.01 mg/kg of detomidine (Detonervin, Ecuphar Italia S.r.l., Milano (MI), Italy); general anesthesia was induced with a dose of 5 mg/kg of tiletamine/zolazepam (Zoletil 50/50, Virbac Italia, Milano (MI), Italy). The sedative and anesthetic medication administered was sufficient to maintain an adequate level of anesthesia for the entire duration of the procedure (which did not exceed 5–10 min in all cases starting from the first incision of the pre-scrotal area and ending with the excision of the second testis). After sedation and the induction of anesthesia, animals were positioned in dorsal recumbency, and the surgical site was prepared, allowing some time for deep anesthesia to initiate. This was done by first washing the whole area with antibacterial soap (DOC SCRUB PVP-IODIO, GARDENING S.r.l., Genova (GE), Italy) to eliminate large contaminant particles, followed by scrubbing with povidone-iodine solution (10%) for a few minutes. The testis was pushed cranially into the inguinal area, and a variable dose of 5–20 mL (according to the length of the incision) of 2% lidocaine solution was injected subcutaneously (SC) along the incisional line to achieve local anesthesia. In no instance did the dose of lidocaine exceed the dose of 15 mg/kg [8].

### 2.3. Surgery

All boars were castrated by open castration. A 4- to 10-cm longitudinal incision was made with a scalpel over the skin covering the inguinally displaced testis. The incision was continued through the underlying tissue, and the parietal tunica vaginalis was incised. The testis was freed from its attachment to the tunica vaginalis by digitally breaking the ligament of the tail of the epididymis, and the spermatic cord was manually stripped from the parietal tunic as proximally as possible to the inguinal ring. According to the treatment assigned to each animal, the surgeon performed either one of the following techniques.

#### 2.3.1. Classical Castration (Group 1)

Animals assigned to group 1 (n = 91) were castrated using the previously described spermatic cord ligation technique and underwent surgical procedures comparable to those currently described in the scientific literature [9] (with the exception of the decubitus position: dorsal instead of lateral). Castration was performed by placing a ligature around the vascular portion of the spermatic cord and the vas deferens using absorbable polyglactin 910 sutures. The suture USP size used was determined according to the respective weight of each boar, with USP 2 (Vicryl Plus, Ethicon Inc., Lamonea Medical Products, Ancona (AN), Italy) being used for animals over 100 kg, USP 1 for animals between 50 and 100 kg, and USP 0 for animals under 50 kg. The ligature was placed proximally to the spermatic cord; the spermatic cord was then transected distally of the ligatures and was inspected for the presence of hemorrhage before being placed back into the tunica vaginalis. Finally, coagulated blood particles were removed from the scrotal pouch, and the surgeon proceeded to castrate the contralateral testis using the same technique described above. 

#### 2.3.2. Tensile Strength Measurements

The age and weight of each animal were recorded prior to the surgery. After the excision of the testis, the diameters of the deferens duct (in mm) of the left and right testis of each animal in both groups were measured at the transection level using grid paper. Additionally, the tensile strength (N) of the left and right deferens ducts was measured using a KOP 24387 (KeenOptics, London, UK) digital dynamometer. For that purpose, a portion of the excised deferens duct was sampled. A standardized length of 4 cm for boars less than 9 months and 5 cm for boars older than 8 months of age (measured with grid paper) was used. Each end of the sampled vas deferens portion was clamped with a straight Klemmer haemostatic forceps. One end was attached to the dynamometer, while the second end was gradually pulled on with increasing force until the deferens duct tore. This procedure was performed by the same individual during the whole study in order to eliminate the risk of error arising from interindividual variability. The tensile strength of each deferens duct was recorded (Newtons). After determining the force necessary to tie a knot between the spermatic funiculus and the ductus deferens without rupturing the latter, the surgeon proceeded with the sutureless technique in the experimental group, as described in the following chapter.

#### 2.3.3. Sutureless Castration (Group 2)

Animals assigned to group 2 (n = 20) were castrated by applying the novel sutureless technique. Prior to the surgery, the age and weight of each animal were recorded. Following extraction of the testis from the vaginal process and stripping of the spermatic cord from the parietal tunic up to the level of the inguinal ring, the deferens duct was manually detached from the tail of the epididymis and freed from the testis spermatic cord. The latter was then divided into a spermatic portion comprised of the deferens duct and a vascular portion consisting of the testicular artery and vein. These two portions were then tied together with 4 to 6 simple knots. The first knot was set as proximally as possible to the internal inguinal ring, while the following knots were placed about 2 cm distally from the first. The force applied to tie the knots was calibrated using the tensile strength results obtained from group 1. After tying an appropriate number of knots, both portions of the spermatic cord were aligned and transected distally from the knots, and the remaining part of the spermatic cord was inspected for hemorrhage before placing it back within the tunica vaginalis. If continuous dripping of blood was observed from the remnant spermatic cord, a double ligature was placed, and the sutureless technique was considered unsuccessful in that particular case. In addition, accidental rupture of the funiculus before the placement of a sufficient number of knots, and thus before adequate haemostasis was achieved, was also deemed a surgical failure. Similarly to the preceding group, the deferens duct was collected and measured to assess tensile strength. Finally, after the removal of coagulated blood particles, the surgeon proceeded to castrate the contralateral testis using the same technique as described above. The different steps applied in the sutureless castration technique are illustrated in Figure 2. 

In both groups, the surgical incision was left open to allow for second-intention healing, providing better drainage of the surgical site in the postsurgical period. The site of the incision and the surrounding area were disinfected with povidone-iodine and sprayed with antibacterial chlortetracycline hydrochloride spray (CycloSpray, Dechra Veterinary Products S.r.l). Prophylactic systemic antibiotic treatment was given in the form of an intramuscular (IM) injection of 30 mg/kg of long-acting oxytetracycline (ENGEMICINA D.D., MSD, MSD Italia S.r.l., Roma (RM), Italy). In addition, a subcutaneous (SC) administration of 300 μg/kg ivermectin (Ivomec., Boehringer Ingelheim Animal Health Italia S.p.A., Milano (MI), Italy) was performed in order to prevent any parasitic complications. 

#### 2.3.4. Follow-Up

At the conclusion of the surgical operation, a veterinarian observed each animal’s awakening until it achieved a stable quadrupedal station. The next day, each farmer was called by telephone to obtain information on the period between recovery and resuming feeding.

### 2.4. Statistical Analysis

All data were entered in an Excel sheet. The Shapiro–Wilk test was used to verify the normal distribution of the data. The mean and standard deviation of each variable for each group was calculated using Stata 17 software (StataCorp, College Station, TX, USA). The correlation between age and tensile strength (CA-T) and weight and tensile strength (CW-T) was calculated using the Pearson linear correlation coefficient R. Results were considered significantly different for a *p*-value lower than 0.05.

## 3. Results

All 111 animals included in this study were successfully castrated through either classical or sutureless castration. No boars from group 2 required the sutureless procedure to be replaced by conventional castration. No complications were recorded during the surgery or the immediate postsurgical period in both groups. 

The age of boars from group 1 ranged between 5 and 36 months, with an average of 18.22 months (sd.10.65), and the age of boars ranged between 4 and 60 months with an average of 17.97 months (sd. 12.80) in group 2. Further results, including weight and the tensile strength and diameter of the deferens duct are shown in Table 1. 

### Correlation 

The castration technique was not taken into account during the evaluation of the relationship between tensile strength and age/weight, and thus all animals were considered part of a single group. In fact, since the variable of interest was the tensile strength of the vas deferens, the observational unit became the vas deferens instead of the boar itself, and therefore each deferens duct (i.e., left or right) was considered as a single observation. Accordingly, the average and median tensile strength, including standard deviation and confidence intervals, were calculated for the totality of the animals (Table 1). Pearson’s linear correlation was used to determine any possible correlation between the deferens duct’s tensile strength and the age and weight of each animal. The correlation coefficient R for tensile strength and age was very close to 1, with R = 0.99 with *p* < 0.05; a strong correlation between weight and tensile strength was found as well (R = 0.96) (Figure 3).

## 4. Discussion

Sutureless orchiectomy is widely applied for the castration of tomcats and was also reported to be successful in young lambs of 4 to 6 days old [10] and pediatric and juvenile dogs [11]. This technique has the potential to be less expensive than traditional orchiectomy and to result in fewer post-operative complications, such as infections [12,13,14]. Another advantage of the sutureless procedure is the duration of surgery, which is significantly shorter. Various castration methods for piglets are described in the current scientific literature [15,16], while the understanding of surgical castration techniques for adult boars is lacking. The prescrotal approach employed in this study was actually found to be poorly described for adult boars. In fact, a scrotal incision is most commonly adopted, with the incision being made ventrally in order to allow for proper drainage [17]. The use of a prescrotal incision for the orchiectomy of adult boars has only been described once, and this approach is thought to allow even better drainage, given the ventral positioning of the incision [9]. Possible complications arising from applying a prescrotal incision include a higher risk of evisceration in the case of an inguinal hernia and the risk of a surgical wound infection developing into peritonitis given the close proximity of the surgical site to the inguinal canal, which can act as a route for ascending infections [18]. The purpose of this study was to assess the feasibility and validity of sutureless orchiectomy in adult boars, and the main objective was to successfully perform this procedure. All 20 animals within group 2 were successfully castrated using the sutureless method without any postsurgical complications or death (same as for group 1: classical castration). The most critical aspect of applying the sutureless technique proved to be the tightening of simple knots between the spermatic and vascular portions of the spermatic cord. At this stage, the vas deferens may break, leading to surgical failure. For this purpose, it was decided to evaluate several variables of the deferens duct during orchiectomies performed in group 1, including their diameter and tensile strength. The latter was correlated with age and weight of the animals in order to determine the maximum traction force to be applied for each boar. 

Although it has been hypothesized that deferens duct tensile strength is directly proportional to age and weight, to the best of our knowledge, no studies correlating these variables have previously been conducted in pigs. In this study, with a relatively small sample size and variable age and weight, a more homogeneous sample was considered by conducting the correlation analysis on a pooled sample. As expected, a strong correlation was found between age and tensile strength (R = 0.99 with *p* < 0.05), as well as weight and tensile strength (R = 0.96 with *p* < 0.05) (Figure 3)**.** According to these results, tensile strength is minimal in young boars and increases progressively with age and weight. Thanks to the identification of these correlations, successful sutureless castration (100% success rate) was performed on 20 boars, calibrating the force needed for optimal nodal tightness without inducing vas deferens rupture.

No post-operative complications were recorded in either group, and animals resumed feeding 5 h after complete recovery from anesthesia. The next step in the validation of sutureless orchiectomy in adult boars will be to perform short-, medium- and long-term follow-ups of animals. 

## 5. Conclusions

Due to its potential advantages, including being simple to execute and master, the sutureless technique can potentially be considered a valid method for adult boar orchiectomy. 

The most critical step in sutureless castration corresponds to the execution of the nodes between the ductus deferens and the spermatic funiculus in order to achieve the proper haemostasis of the vessels. The force required to tie these knots is directly proportional to the age and weight of the animals. This study has laid the cornerstone needed to proceed with large-scale clinical studies on the sutureless castration of adult boars. Future research is needed in order to explore other essential aspects of this technique, such as wound healing, postsurgical recovery, and the occurrence of short and long-term complications. An in-depth investigation of the occurrence of postsurgical complications was left to the next stage of research, which will consist of a large-scale clinical study based on the findings of this study.

## Figures and Tables

**Figure 1 animals-13-00407-f001:**
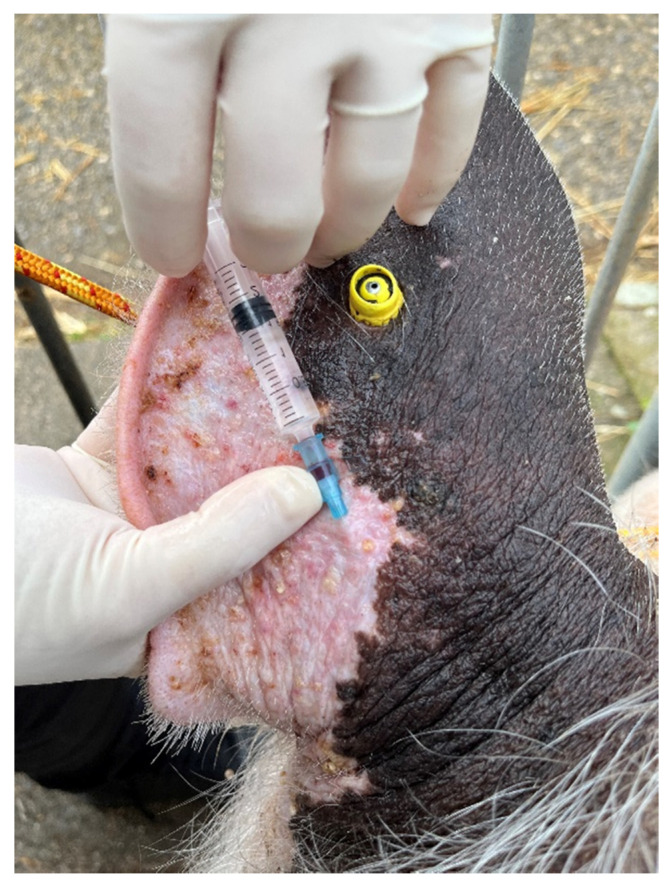
IV access for the administration of anesthetic drugs was achieved through the auricular veins.

**Figure 2 animals-13-00407-f002:**
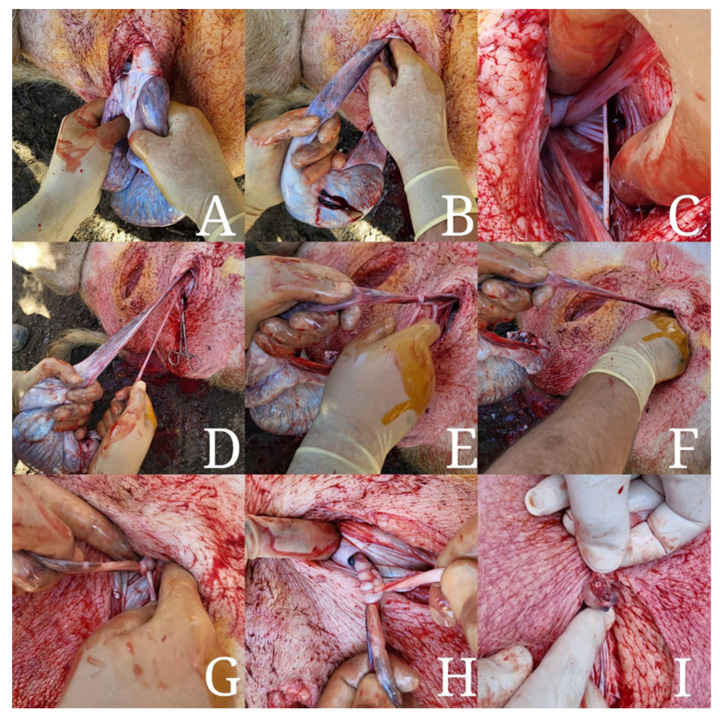
Illustration of orchiectomy using the sutureless technique: Initiate by making a skin incision in the prescrotal area with a scalpel blade. Continue the incision through the underlying tissue and parietal tunica vaginalis and exteriorize the testis (**A**). Digitally rupture the ligament of the tail of the epididymis and strip the spermatic cord from the tunica vaginalis up to the level of the inguinal canal (**B**,**C**) before manually detaching the deferens duct from the epididymis, splitting the spermatic cord into a spermatic and vascular portion (**D**). Tie both portions together using 4 to 6 simple knots (**E**,**F**). Place the first knot as proximally as possible to the internal inguinal ring (**G**) and the following knots about 2 cm distally to the first (**H**). Finally, align the spermatic and vascular portions of the spermatic cord and transect using a scalpel blade No 22. Assess that correct haemostasis of blood vessels is achieved (**I**).

**Figure 3 animals-13-00407-f003:**
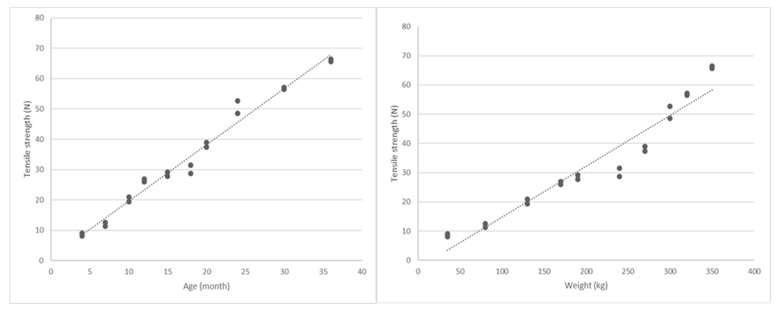
The left graph shows the correlation between age and tensile strength in the pooled sample; the right graph shows the correlation between weight and tensile strength in the pooled sample. A significantly strong correlation was found (R = 0.99 and R = 0.96, respectively (*p* < 0.05)).

**Table 1 animals-13-00407-t001:** Average and standard deviation of variables recorded for groups 1 and 2; R: right; L: left; DD: deferens duct.

Group 1	Age (months)	Weight (kg)	Tensile Strength R (kg)	Tensile Strength L (kg)	Right DD Diameter	Left DD Diameter	DD Tensile Strength R (N)	DD Tensile Strength L (N)
Average	18.22	211.75	2.35	2.46	3.67	3.64	22.42	23.76
standard deviation	10.65	78.90	1.16	1.16	1.72	1.71	10.42	10.20
Group 2								
Average	17.97	192.72	2.22	2.40	4.05	4.02	22.17	23.56
standard deviation	12.80	93.85	1.14	1.30	1.62	1.67	10.83	12.75

## Data Availability

Not applicable.

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
