# Peer review of "Sutureless Technique for Surgical Castration in Adult Boars: A Feasibility Study"

_animals, 2023, doi:10.3390/ani13030407_

Round 1
Reviewer 1 Report
A brief summary
Adult boars need to be castrated to prevent the risk for boar taint in meat, making the meat unfit for consumption. This study describes a novel sutureless method of surgical castration, orchiectomy. The tensile strength of the deferent ducts were measured in order to use estimate the force needed to tighten the knots used to stop the bleeding after orchiectomy. Then the new technique was successfully applied for castrating adult boars. The method is well described and all necessary details are included. This is a new method, which is technically simple and potentially causes less post-surgical complications. The study opens up the possibility for further studies, which could ultimately lead to a new, better method for castration adult boars.
General Comments
I mention this also in Specific Comments; see my comments to lines 264-266. There is no data in the paper on the boras being observed after the castration. The authors mention though that no complications were recorded “in the immediate postsurgical period” (line 199), no “post chirurgical complications or eventual death” (line 242-243), and “the animals resumed feeding 2-5 hours after surgery” (line 264-265). I understand that not real follow up has been done, but this needs to be clarified. It appears that the boars were observed for at least 2-5 hours. Were all boars observed until they resumed feeing? What does “eventual death” mean? I suggest specifying in the manuscript how long the boars were observed after surgery, what kind of observations were made and results of those observations. It is important for the conclusions. In addition, some more details regarding the animals used and the where the castrations were performed are needed, see Specific Comments to Materials and Methods. 2.1.
Specific Comments
Line 17. The sentence: “Due to testosterone and its metabolites…” is not entirely correct. As the authors correctly explain later in the manuscript, the taint in meat, called boar taint, in older boars is often caused by androstenone. Androstenone is a testicular steroid but cannot be not described as a metabolite of testosterone. The presence of testosterone does not make the meat unfit for consumption, as it is implied also in line 51. Please correct both sentences.
Line 19. “To date, literature concerning surgical orchiectomy in boars is limited “. I recommend “To date, literature concerning surgical orchiectomy in adult boars is limited “. Castration of young boars has been well studied.
Line 24. The “group 1” has not really been previously defined. I t would probably be better to write the “first group” instead.
Materials and Methods. 2.1. Please explain the origin of the animals. Did you select the animals in any way? Were the castrations performed on the farms or in a clinic? I assume the investigators went to the farms responding to the calls from the farmers but it is not clear.
Line 83. I assume the authors mean Figure 1. Please correct.
Line 50. The world “career” does not seem to be appropriate. Please consider another wording.
Line 127-129. The last sentence within this paragraph is repeated. Delete.
Line 130- 145. This paragraph should be a separate section with a separate heading that refers to tensil strength of the vas deferens measurements because this paragraph applies to both group 1 and group 2. The current heading focuses on group 1 only.
Line 168 – I recommend starting a new paragraph beginning with “In both groups…”, or possibly, a new section with a separate heading.
Line 174 – Use of oxytertracycline which is a broad spectrum antibiotic with long withdrawal time seems to be excessive. The same could apply for ivermectin. In my part of Europe, the Nordic countries, such treatment after castration would not be recommended. However, I do not know the local conditions, so this is just a comment and the authors do not need to respond.
Line 176 – 177. The sentence “Similar to the preceding…” implies that the described directly above post-surgery treatment was applied only to group 2. When I read it first, I thought (and I still think) that that description is true for both groups. Please make it clearer. If the latter is true, the last sentence might not be necessary. The measurements of the deferens ducts were previously described for both groups, lines 130-145.
Table 1. The table does not look good in my uploaded file, might need some editing. Follow the journal/editor recommendation.
Line 213. I assume it should be “Table 1”. Please correct.
Line 231. I am not sure in the word “later” is needed her. Consider deleting.
Line 264 -266. It is never mentioned in the manuscript whether and how long the animals were observed after surgery. Was it only feeding that was recorded? On the farm or at the clinic? – see also my comment to M&M point 2.1 above. That should be described in Materials and Methods and the results of the observations should be included in the Results section
Conclusions. The authors state that in the first sentence that the sutureless technique can be considered a valid technique . Only later do they explain that the technique needs to be evaluated further. I recommend to state that already in the first sentence. In my opinion, the short conclusion from the study is that the technique can potentially be considered a valid technique, however further studies are required. In line 280, it is stated that “it would be interesting” to validate the technique. I strongly recommend rewording this statement; further studies are necessary.
Line 269. “Although it was not included in the study… .” Not well formulated sentence. What was not included?
Line 270. The sentence: “This research was carried ….”. It is of course interesting to look at the “students’ learning curve”, however I fail see that this is a conclusion from this particular work. Those aspects, i.e. whether the technique is easy to master, as mentioned by the authors, were not a subject of the study and therefore should not be discussed to that extend in the Conclusions section. I suggest reducing or deleting this part of conclusions.
Author Response
Please see the attachement

Reviewer 2 Report
The study design is simple and well presented, I appreciated this.
The description of the surgical procedure was the goal of the paper, for me, indeed its presentation is well described and clear. The English is understandable for me but I am not a native speaker so I refrained from comments on the subject.
The statistics section in my opinion should be better presented
I only have a few suggestions that I think could make some small improvements to the paper.

Author Response
Please see the attachement
